# The First *Pseudomonas* Phage vB_PseuGesM_254 Active against Proteolytic *Pseudomonas gessardii* Strains

**DOI:** 10.3390/v16101561

**Published:** 2024-09-30

**Authors:** Vera Morozova, Igor Babkin, Alina Mogileva, Yuliya Kozlova, Artem Tikunov, Alevtina Bardasheva, Valeria Fedorets, Elena Zhirakovskaya, Tatiana Ushakova, Nina Tikunova

**Affiliations:** 1Institute of Chemical Biology and Fundamental Medicine Siberian Branch of Russian Academy of Sciences, Novosibirsk 630090, Russia; morozova@niboch.nsc.ru (V.M.); a.mogileva@g.nsu.ru (A.M.);; 2Faculty of Natural Sciences, Novosibirsk State University, Novosibirsk 630090, Russia

**Keywords:** *Pseudomonas gessardii*, environmental *P. fluorescens* complex, *Pseudomonas bacteriophage*, proteolytic activity, myovirus, genome inversion

## Abstract

Bacteria of the *Pseudomonas* genus, including the *Pseudomonas gessardii* subgroup, play an important role in the environmental microbial communities. Psychrotolerant isolates of *P. gessardii* can produce thermostable proteases and lipases. When contaminating refrigerated raw milk, these bacteria spoil it by producing enzymes resistant to pasteurization. One possible way to prevent spoilage of raw milk is to use *Pseudomonas* lytic phages specific to undesirable *P. gessardii* isolates. The first phage, *Pseudomonas* vB_PseuGesM_254, was isolated and characterized, which is active against several proteolytic *P. gessardii* strains. This lytic myophage can infect and lyse its host strain at 24 °C and at low temperature (8 °C); so, it has the potential to prevent contamination of raw milk. The vB_PseuGesM_254 genome, 95,072 bp, shows a low level of intergenomic similarity with the genomes of known phages. Comparative proteomic ViPTree analysis indicated that vB_PseuGesM_254 is associated with a large group of *Pseudomonas* phages that are members of the *Skurskavirinae* and *Gorskivirinae* subfamilies and the *Nankokuvirus* genus. The alignment constructed using ViPTree shows that the vB_PseuGesM_254 genome has a large inversion between ~53,100 and ~70,700 bp, which is possibly a distinctive feature of a new taxonomic unit within this large group of *Pseudomonas* phages.

## 1. Introduction

The genus *Pseudomonas* (family *Pseudomonadaceae*) contains Gram-negative, motile, non-spore-forming, catalase- and oxidase-positive gamma-proteobacteria. Currently, there are more than 300 validly published species in this genus (https://lpsn.dsmz.de/genus/pseudomonas, accessed on 3 June 2024) and they are divided into three large lineages, namely *P. fluorescens*, *P. pertucinogenes*, and *P. aeruginosa* lineages [1,2]. In the *P. fluorescens* lineage, the most diverse and numerous is the *P. fluorescens* complex, which is subdivided into nine phylogenetically different subgroups. These subgroups are named according to their prototype members: *P. protegens*, *P. chlororaphis*, *P. corrugate*, *P. koreensis*, *P. jessenii*, *P. mandelii*, *P. fragi*, *P. gessardii*, and *P. fluorescens* [3].

Members of the *Pseudomonas* genus exhibit great metabolic diversity and are therefore able to colonize a wide range of natural habitats [4,5,6,7]. Some human and animal pathogens belong to this genus, and the most famous and studied of them is *Pseudomonas aeruginosa* [8,9]. Due to their wide variability in metabolic activities, a number of *Pseudomonas* species are producers of various compounds with anti-bacterial and antiviral properties [10,11,12]; some of them can be used as oil destructors [13,14] or producers of organic substances for industry [15,16,17]. The genus includes both phytopathogens (*P. syringae*) and, on the contrary, phytoprotective *P. protegens* and *P. chlororaphis* species. Phytoprotective *Pseudomonas* spp. ensure the stability of the rhizosphere and compete with pathogenic microorganisms by producing anti-fungal and anti-bacterial metabolites [10,11,18,19,20]. Since *Pseudomonas* spp. play an important role in the ecosystem, their bacteriophages can influence the composition of microbial communities, thereby maintaining the stability of microbial ecosystems. In particular, lytic phages can destroy the cells of plant commensal *P. protegens*. At the same time, they can also be used as non-chemical antimicrobials to kill plant pathogens, such as bacteria from the *P. syringae* and *P. fluorescens* groups [21].

In addition, it has been shown that some psychrotolerant members of the *Pseudomonas* genus, namely *P. gessardii*, *P. fluorescens*, *P. fragii*, and some others, can produce thermostable proteases and lipases [22,23]. Therefore, these bacteria pose a problem for the dairy industry by contaminating refrigerated raw milk and producing these enzymes, which are resistant to pasteurization and spoil pasteurized milk during its storage [23,24,25]. One possible way to prevent the spoilage of raw milk is to use *Pseudomonas* lytic phages specific to these undesirable *Pseudomonas* species [26,27,28]. The search for *Pseudomonas* phage complete genomes in the NCBI GenBank (https://www.ncbi.nlm.nih.gov/genbank, accessed on 1 July 2024) reveals more than 1100 phage genomes, of which 850 were isolated using pathogenic *P. aeruginosa* as a bacterial host. Apparently, *P. aeruginosa* phages have been identified as potential therapeutic agents, and their development is being pursued due to the increasing antibiotic resistance of this organism and its importance in clinical practice. Among the remaining 250 *Pseudomonas* phages, 167 are specific to *P. syringae*, whereas only 37 phages are specific to bacteria from the *P. fluorescens* group. Although the genomes and biological properties of several *P. fluorescens* phages have been studied [29,30,31,32,33,34,35,36], much remains to be learned about these viruses and their interactions with their hosts. Notably, no phages specific to the *P. gessardii* subgroup have been isolated and characterized (Appendix A).

Here we describe the first *Pseudomonas* phage vB_PseuGesM_254, which is specific to the environmental proteolytic *P. gessardii* isolate. Biological properties, host range, and phage complete genome were analyzed, and putative taxonomy position of the phage was proposed.

## 2. Materials and Methods

### 2.1. Bacterial Strain Isolation, Identification, and Growth Conditions

All *Pseudomonas* strains used in the study were obtained from the Collection of Extremophilic Microorganisms and Type Cultures (CEMTC) of the Institute of Chemical Biology and Fundamental Medicine SB RAS, Novosibirsk, Russia (Appendix A). The host strain *Pseudomonas* CEMTC 4637 was isolated from a sediment sample, taken from the Chemal River in the Altai Mountains, Russian Siberia. One gram of the sediment sample was suspended in 10 mL of sterile phosphate buffered saline, PBS (5.84 g of NaCl, 4.72 g of Na_2_HPO_4_, and 2.64 g of NaH_2_PO_4_×2H_2_O per 1 L, pH 7.5), and agitated using a Vortex mixer. Insoluble particles were pelleted via low-speed centrifugation (1000× *g*, 5 min at room temperature, RT), and the resulting supernatant was used to isolate bacteria. Nutrient agar plates (Microgen, Obolensk, Russia) were inoculated with tenfold dilutions of the supernatant, which were prepared in sterile PBS (pH 7.5). The plates were incubated at 25 °C for 20 h. After that, they were checked for the presence of bacterial colonies. To obtain pure strains, the grown bacterial colonies were streaked on the nutrient agar plates and incubated again at 25 °C for another 20 h. This step was repeated twice. Finally, samples of the bacterial culture were spread on another nutrient agar plate to confirm its purity.

To identify a bacterial isolate, a fragment of the 16S rRNA gene (1308 bp) was sequenced, as described previously [37]. The taxonomy of the strains was clarified using phylogenetic analysis of concatenated nucleotide sequences of the *rpoB*, *gyrB*, and *rpoD* genes as described previously [10]. The primer sequences, their annealing temperature, and the calculated sizes of the PCR products are listed in Appendix A.

To prepare bacterial DNA template for PCR, a single bacterial colony was picked, placed in 10 µL of sterile deionized water, and suspended. The resulting bacterial suspension was heated for 5 min at 95 °C, and the bacterial debris was removed via centrifugation (10,000× *g*, 5 min, RT). An aliquot of the supernatant (1 µL) was used as a template for amplification. PCR products were purified using 0.6% SeaKem GTG-agarose gel electrophoresis (Lonza, ME, USA) and then sequenced using the Sanger method with the BigDye Terminator v.3.1 cycle sequencing kit and an ABI 3500 genetic analyzer (Applied Biosystems, Foster City, CA, USA). The obtained data were analyzed using the FinchTV 1.4.0 tool (https://finchtv.software.informer.com, accessed on 7 May 2024). Partially overlapping fragments were assembled using the SeqMan tool from the Lasergene Evolution Suite (DNASTAR, Madison, WI, USA). The obtained nucleotide sequences were compared with reference sequences from the NCBI GenBank database. The species was routinely determined based on the criterion that the sequences of the 16S rRNA gene must be more than 98% identical to the reference sequence.

To clarify the taxonomy of a number of *Pseudomonas* isolates, the PCR products corresponding to the partial sequences of *rpoB*, *gyrB*, and *rpoD* genes were sequenced using the same primers, which were used for amplification (Appendix A). The obtained data were analyzed and assembled in the same manner used for the 16S rRNA gene. The obtained sequences were compared with the RefSeq Reference Genome Database of the NCBI GenBank (search was limited to *Pseudomonas* genus, taxid:286 AND Type strains). The closest gene sequences were downloaded, combined into a concatenated sequence for each reference genome, and used for phylogenetic analysis (Appendix A). Phylogenetic trees were constructed using the MEGA 11.0 tool [38], using the ClustalW algorithm to align concatenated sequences and the maximum likelihood method to obtain trees.

To determine the optimal growth temperature for the *Pseudomonas* CEMTC 4637 strain, its growth was examined in 2% nutrient broth (Microgen, Obolensk, Russia) at various temperatures (10 °C, 20 °C, 25 °C, 30 °C, and 37 °C) over a 36-h period. As a result, the *Pseudomonas* CEMTC 4637 strain was routinely cultured at 25 °C either in nutrient broth or on nutrient agar plates.

The strain was deposited in CEMTC of the Institute of Chemical Biology and Fundamental Medicine SB RAS, Novosibirsk, Russia. Bacterial suspension of CEMTC 4637 strain was stored at a temperature of −80 °C in tubes containing LB medium (tryptone 1% and yeast extract 0.5%, sterile glycerol 25%).

In order to determine the host range of the phage under study, a number of environmental *Pseudomonas* strains from CEMTC were used (Appendix A). Strains of *Pseudomonas* that are susceptible to phage PseuGes_254 were tested for their proteolytic activity by spreading them on milk agar plates (skim milk 1.0%, peptone 0.1%, NaCl 0.5%, agar 2.0%) and incubating plates at 25 °C for 18 h. The formation of clear, transparent zones around the bacterial colonies confirmed the production of proteases.

### 2.2. Phage Isolation and Propagation

The *P. gessardii* phage vB_PseuGesM_254 (hereinafter PseuGes_254) was obtained from the same sediment sample that was used to isolate its bacterial host. The supernatant obtained after low-speed centrifugation of the sediment sample was filtered through a 0.22 µM pore size membrane filter (Millipore, Guyancourt, France) and stored at 4 °C. Later, it was screened for phages specific to the *P. gessardii* CEMTC 4637 strain using the double-agar method as described previously [27]. In brief, a fresh layer of *P. gessardii* CEMTC 4637 strain in the top agar (tryptone 1%,yeast extract 0.5%, agar 0.8%) was prepared, and 10 µL aliquots of sterilized supernatant were spotted on it. The plates were then incubated at 25 °C overnight and examined for phage-formed plaques; the resulting individual plaques were cut out and placed in 100 µL of sterile PBS, then the mixture was incubated with shaking for 16 h at RT to extract phage particles. Then, tenfold dilutions of the phage suspensions were dripped onto the top agar containing the *P. gessardii* CEMTC 4637 cells, and plates were incubated overnight at 25 °C to obtain single phage plaques for subsequent phage extraction. The cycle of phage plating and extraction was repeated three times.

To propagate the obtained PseuGes_254 phage, *P. gessardii* CEMTC 4637 cells were grown in 2% nutrient broth with shaking at 25 °C. One hundred microliter aliquots of bacterial suspension were taken every 20 min, and OD_600_ was measured using a spectrophotometer (SmartSpecPlus, BioRad, Hercules, CA, USA). PseuGes_254 was added to the exponentially growing *P. gessardii* CEMTC 4637 culture (OD_600_ = 0.4) at a multiplicity of infection (MOI, i.e., the ratio of phage to bacterium) of 0.1. The infected culture was shaken at 25 °C for two hours until cell lysis was observed. Afterwards, bacterial debris and cells were removed using centrifugation (10,000× *g*, 15 min, RT), and viral particles were precipitated from the obtained supernatant using polyethylene glycol 6000 (AppliChem, Darmstadt, Germany) and 2.5 M NaCl, as described previously [39].

### 2.3. Biological Properties and Host Range Assay

Phage adsorption assays and burst size experiments were performed according to [40,41], with slight modifications, including incubation of bacterial cultures at 25 °C. To estimate the phage adsorption rate, *P. gessardii* CEMTC 4637 cells were grown at 25 °C with shaking until the OD600 reached 0.4. Bacterial cells were mixed with PseuGes_254 particles at a MOI of 0.001 plaque forming units per cell (pfu/cell). The phage–cell mixture was incubated for 15 min at 25 °C without shaking. Every minute, 100 μL aliquots were taken from the mixture, and the phage titer in each aliquot was determined. The PseuGes_254 adsorption curve was constructed based on the obtained data. To evaluate phage burst size, the PseuGes_254 phage particles were added to the exponentially growing *P. gessardii* CEMTC 4637 host strain at an MOI of 0.01 pfu/cell, and the phage–cell mixture was incubated for 5 min at 25 °C for phage adsorption. Then, the cells were pelleted by centrifugation (4000× *g*, 10 min, RT), re-suspended in 10 mL of 2% nutrient broth, and incubated for 70 min at 25 °C with shaking. Culture aliquots were collected every 5 min, and the phage titer was determined. A lytic activity assay of PseuGes_254 was performed as described previously [42] with some modifications. Bacterial cultures of the *P. gessardii* CEMTC 4637 strain were grown with shaking in 2% nutrient broth at 10 °C and 25 °C, respectively, until the OD_600_ reached approximately 1.0. Both cultures then were infected with phage at a MOI of 0.01 pfu/cell. The phage–cell mixtures were further incubated with shaking at 10 °C and 25 °C respectively, for 20 h. During the first seven hours, 100 μL aliquots were taken from each culture every hour. The last aliquot was collected after 20 h of incubation. Cell titers were determined in these aliquots by diluting them and plating the dilutions on nutrient agar. Inoculated plates were incubated overnight at 25 °C. The next day, bacterial colonies were counted, and the titer of bacterial cells was calculated. The multistep bacterial killing curve for PseuGes_254 was constructed using the obtained data. *P. gessardii* CEMTC 4637 cultures incubated under the same conditions without phage were used as a control of bacterial growth. The host range for PseuGes_254 was determined using a spot-assay method [43]. Strains of the *P. gessardii* subgroup from the Collection of EMTC were tested for susceptibility to the phage.

All experiments on the biological properties of PseuGes_254 were performed twice, three times in each repeat. The statistical analysis and graphs were prepared using GraphPad Prizm v. 8.0 (https://www.graphpad.com, accessed on 14 May 2024).

### 2.4. Phage Plaques and Phage Particles Morphology

The morphology of phage plaques was determined in the layer of the *P. gessardii* CEMTC 4637 host culture in the top agar. PseuGes_254 was added to the bacterial culture in a low titer (10^3^ pfu/mL) to obtain single plaques. The plate was incubated overnight at 25 °C, and the morphology of phage plaques was studied. The method of electron microscopy with negative staining was used to visualize PseuGes_254 particles. The phage suspension (~10^8^ pfu/mL) was adsorbed on a copper grid and contrasted on a drop of 1% uranyl acetate; afterwards, the grid was examined for viral particles using a JEM 1400 transmission electron microscope (JEOL, Tokyo, Japan). The dimensions of the head and tail were calculated from ten independent measurements of phage particles in micrographs.

### 2.5. Complete Genome Sequencing and Analysis

The PseuGes_254 DNA was obtained as described previously [44]. Briefly, phage particles were precipitated from bacterial lysate using PEG 6000 (AppliChem, Darmstadt, Germany) supplemented with 2.5 M NaCl. Precipitate was pelleted by centrifugation, and then the pellet was dissolved in STM buffer, containing 10 mM NaCl, 50 mM Tris-HCl, and 1 mM MgCl_2_ (pH 8.0). Further, the phage preparation was incubated with 5 µg/mL of RNase and DNase (Thermo Fisher Scientific, Waltham, MA, USA) for 1 h at 37 °C. Next step, SDS, proteinase K (Thermo Fisher Scientific, Waltham, MA, USA), and EDTA were added to final concentrations of 0.5%, 100–200 mkg/mL, and 20 mM, respectively, and the mixture was incubated for 3 h at 55 °C. At the final stage, phenol–chloroform DNA extraction was performed, and the DNA from the phage was precipitated by adding 2.5 volumes of 96% ethanol. The Nextera DNA Sample Preparation Kit (Illumina, Inc., San Diego, CA, USA) was applied to prepare a paired-end genome library according to the manufacturer’s instructions, and sequencing was performed using the MiSeq Benchtop Sequencer and MiSeq Reagent Kit v.1 (Illumina Inc., San Diego, CA, USA). The genome was *de novo* assembled using the SPAdes genome assembler v.3.15.2 [45] (https://github.com/ablab/spades, accessed on 16 March 2023). PhageTerm v.1.0.12 software [46] (https://galaxy.pasteur.fr/?tool_id=toolshed.pasteur.fr%2Frepos%2Ffmareuil%2Fphageterm%2FPhageTerm%2F1.0.12&version=1.0.12&__identifer=q05ix39bvq, accessed on 8 August 2023) was applied to determine the position of the phage PseuGes_254 termini. Rapid Annotation Subsystem Technology (RAST) v.2.0 [47] (https://rast.nmpdr.org, accessed on 12 August 2023) was used to annotate the putative ORFs. The obtained ORFs were verified manually using BLAST algorithms against nucleotide and protein sequences, deposited in the NCBI GenBank (https://ncbi.nlm.nih.gov, accessed on 23 September 2023). In addition, the InterProScan [48], HHPred, and HMMER tools [49] were applied for the identification of hypothetical proteins. BioEdit [50] and MAFFT [51] (https://mafft.cbrc.jp/alignment/server, accessed on 30 September 2023) tools were used for editing and aligning nucleotide sequences. Ori-Finder software [52] (https://tubic.org/Ori-Finder, accessed on 18 June 2024) was used to analyze the PseuGes_254 genome for the presence of putative origin of replication. Comparative genome analysis was carried out using ViPTree [53] (https://www.genome.jp/viptree, accessed on 11 June 2024) and VIRIDIC tools [54] (http://rhea.icbm.uni-oldenburg.de/VIRIDIC, accessed on 15 June 2024). The search for virulence factors and antibiotic resistance genes was carried out using the Virulence Factor (VR) database (http://www.mgc.ac.cn/VFs, accessed on 30 April 2024) and Antibiotic Resistance Gene (ARG) database (https://cge.food.dtu.dk/services/ResFinderFG/, accessed on 30 April 2024), respectively.

The phylogenetic analysis of the essential proteins encoded by the PseuGes_254 genome was carried out as follows: the most similar protein sequences identified by the BLASTP search were extracted from the NCBI Genbank; then, the sequences were aligned and analyzed using MEGA 11.0 [38]. The PseuGes_254 genome was deposited in the NCBI GenBank database (accession number OR575930).

## 3. Results

### 3.1. Phage Particles Morphology, Biological Properties

Electron microscopy indicated that the PseuGes_254 particle consists of a large capsid (Ø 79.2 ± 1.92 nm) connected to a long contractile tail (L = 139.6 ± 5.6 nm). Therefore, the virion morphology corresponds to the myovirus morphotype (Figure 1A).

The phage formed small, clear plaques on the host CEMTC 4637 layer in the top agar. Adsorption assay revealed that 70% of the PseuGes_254 phage particles adsorbed at the host bacterial cells in 12 min (Figure 1B). A one-step growth experiment revealed a latent period of ~50 min with a burst size of ~10 phage particles per infected cell (Figure 1C). A multistep bacterial killing curve for the phage PseuGes_254 showed that the number of live host bacteria decreased by two orders of magnitude two hours after infection at 25 °C and four hours at 10 °C. After that, the titer of bacterial cells began to slowly increase (Figure 1D). Importantly, PseuGes_254 is able to lyse host cells at low temperature, 10 °C (Figure 1D).

### 3.2. Bacterial Hosts

To determine the host range of PseuGes_254, 19 environmental Pseudomonas strains, previously identified by sequencing the 16S rRNA gene as members of the *P. gessardii* subgroup, were used (Appendix A). It has been revealed that the phage infects three Pseudomonas strains, namely CEMTC 4637 (host strain), CEMTC 4644, and CEMTC 5432 (Table 1). Bacterial strains sensitive to PseuGes_254 were isolated from samples collected in the Altai Republic. These samples contained water with sediments from the cold tributaries of the Katun River, which in turn forms the great Siberian Ob River. (Table 1).

All three bacterial strains that were sensitive to PseuGes_254 showed proteolytic activity on plates with milk agar (Appendix A). The temperature range of bacterial growth was determined for the host strain CEMTC 4637, and it turned out to grow in a wide temperature range from 10 °C to 37 °C, with an optimal temperature of 25–30 °C. Slow growth at a temperature of 10 °C was detected (Appendix A), indicating the possibility of the bacterium to multiply in cold raw milk and to spoil it.

To confirm taxonomic classification of Pseudomonas isolates, fragments of the rpoB, rpoD, and gyrB genes (634 bp, 632 bp, and 528 bp, respectively) were sequenced, and phylogenetic analysis of the concatenated sequences (1794 bp) was performed. A number of *P. gessardii* subgroup members (Appendix A), were added to the analysis. As a result, the majority of analyzed bacterial strains formed a group, which includes three strains sensitive to PseuGes_254 and five other bacterial strains isolated from tributaries of the Katun River and Ob River (Figure 2). This group clustered with *P. gessardii* type strains; therefore, PseuGes_254 hosts presumably may represent a subspecies in the *P. gessardii* species.

### 3.3. PseuGes_254 Genome Characteristics

The PseuGes_254 phage genome was sequenced and assembled de novo using the SPAdes genome assembler v.3.15.2. As a result, the circular contig with the average coverage of 216 was obtained. The Phage Term tool revealed that the PseuGes_254 genome was terminally redundant and contained direct terminal repeats (DTRs) with a length of 628 bp; so, the studied phage presumably uses a T7-like strategy for DNA packaging (Appendix A). The genome length is 95,072 bp with a GC content of 43%, in contrast to the GC content of the *P. gessardii* genome, which has been reported to be 58% [55]. A total of 195 genes were identified in the genome; 183 of them encode putative proteins, and the remaining twelve correspond to tRNAs (Figure 3). Functions were predicted for 62 putative proteins based on their amino acid sequences and domain structure similarity. The remaining 121 sequences were defined as hypothetical. No genes responsible for the lysogenic life cycle (integrase, transcription repressor, etc.) were found in the genome; therefore, it is supposed that the phage is lytic. No virulence factors or antibiotic resistance genes were found in the genome.

The genes responsible for the metabolism of nucleic acids are divided into two groups. The first of them contains the genes of DNA polymerase A, primase, ssDNA-binding protein, thymidylate synthase, and alpha/beta subunits of ribonucleoside diphosphate reductase; endonuclease, DNA helicase, tRNA splicing ligase, and DNA ligase are combined into another cluster; these groups are separated by a cluster of genes encoding structural proteins (Figure 3). Gene corresponding to RNA polymerase was not determined; obviously, the phage uses a bacterial transcription machine. Eleven of the twelve tRNA genes are located together between the genes encoding the small and large subunits of terminase. The different GC content in the phage genome and host genome, as well as a large set of phage tRNA genes, suggests that the codon usage of PseuGes_254 differs from that in host cells.

Three putative receptor-binding proteins (tail spike protein and two tail fiber proteins) were identified in the cluster of structural genes (Figure 3), and this fact suggests that PseuGes_254 has a structurally sophisticated adsorption complex.

The AT-rich region at positions 70,649–70,773 nt was determined. It was assumed that this region may be a part of the phage origin of replication, as it was shown previously for bacteria, phages, and plasmids [56]. Therefore, Ori-Finder software [52] was used to analyze the PseuGes_254 genome (https://tubic.org/Ori-Finder, accessed on 18 June 2024), and the putative origin of replication was revealed at positions 70,536–70,927 nt in the genome of PseuGes_254 (Figure 4, Appendix A).

### 3.4. Comparative Genomic Analysis

A search in the GenBank database using the BLASTN algorithm revealed a low similarity between the PseuGes_254 nucleotide sequence and the genomes of other identified phages (query cover ~1% with nucleotide identity, NI, 75–80%). Only two similar nucleotide sequences (46,990 bp and 46,611 bp) were found in the whole-genome shotgun contigs database, and both of them belong to the Pseudomonas helleri isolate WonhQNNPkY_bin.7.MAG (GenBank ID: CALTWU000000000.1), which was obtained during metagenome analysis of human skin microbiom (BioProject ID: PRJEB47281). These contigs, namely NODE_144_length_46990_cov_333.200810 (GenBank ID: CALTWU010000030.1), and NODE_148_length_46611_cov_317.059283 (GenBank ID: CALTWU010000032) were extracted from the GenBank and used along with the PseuGes_254 genome for further comparative genomic analysis (Figure 5).

Alignment of the nucleotide sequence of PseuGes_254 and two contigs from BioProject ID: PRJEB47281 indicated that the NODE_148 sequence is homologous to the PseuGes_254 sequence from 1 to 23,844 bp and from 70,729 to terminal 95,072 bp, whereas the NODE_144 sequence is homologous to the PseuGes_254 genome from 23,853 to 70,728 bp. According to the alignment results, there is a gap between the NODE_148 and NODE_144 of 9 bp, which probably did not allow these two sequences to be combined into a single contig (Figure 5 and Appendix A). It has been suggested that these sequences are fragments of the putative genome of an unknown phage related to PseuGes_254. This sequence from the P. helleri isolate WonhQNNPkY_bin.7.MAG is hereinafter referred to as the putative phage Wonh.

ViPTree analysis revealed that the PseuGes_254 phage together with the putative phage Wonh are associated with a large group of *Pseudomonas* phages, belonging to the *Skurskavirinae* and *Gorskivirinae* subfamilies, and to the *Nankokuvirus* genus. PseuGes_254 and phage Wonh together form a separate group, which is distant from other related *Pseudomonas* myophages (Figure 6).

ViPTree alignment showed that PseuGes_254 and phage Wonh possess in-between gene syntheny and, at the same time, a large inversion in both genomes (region between 53,100 bp and 70,700 bp) was found comparing to other related *Pseudomonas* phages (Figure 7). The inverted region in the PseuGes_254 genome contains genes encoding primase, DNA polymerase A, ssDNA-binding protein, thymidylate synthase, two ribonucleoside reductase subunits, and a number of putative proteins (Figure 3). No repeats or any structures of mobile elements were found at the beginning of this inverted region of the genome. At the same time, AT-repeats, corresponding to the putative origin of replication (Figure 4), were located at the end of this fragment (Figure 3).

The VIRIDIC tool was used to assess the genome similarity between the related Pseudomonas phage genomes (Figure 8). The level of similarity between the PseuGes_254 and the putative phage Wonh was ~66%, which is below the cut-off (70%) for combining phages into one genus [57]. Therefore, the PseuGes_254 may be the only species of the new genus. Other related *Pseudomonas* phages demonstrated a very low level of intergenomic similarity with PseuGes_254 and the putative phage Wonh (Figure 8).

### 3.5. Phylogenetic Analysis of Eessential Phage Proteins

Phylogenetic trees of the PseuGes_254 terminase large subunit and major capsid protein were constructed using the corresponding predicted amino acid sequences (Appendix A). The topology of the resulting phylogenetic trees was similar to that of ViPTree analysis (Figure 6). Both studied amino acid sequences were distant from those of members of the Skurskavirinae and Gorskivirinae subfamilies and Nankokuvirus genus (Appendix A).

## 4. Discussion

A phage specific to *P. gessardii* bacteria was isolated for the first time. The study of the biological properties of this phage, PseuGes_254, indicated that it has lytic activity and can infect several *P. gessardii* isolates. It is worth noting that both the phage and sensitive bacterial strains were isolated from water samples collected in cold rivers from the Altai Republic. This may indicate a limited distribution area for this phage, or *P. gessardii* strains from this region have some special features that make them suitable hosts for this phage, and they co-evolve with it. To confirm or reject this hypothesis, detailed analysis of the genomes and enzymatic activity of the susceptible *P. gessardii* strains is required.

Sequences of the PseuGes_254 genome and most of the proteins produced by it substantially differ from those of other phages available in the GenBank database. The closest sequences to the PseuGes_254 genome belong to two contigs found in the *Pseudomonas helleri* WonhQNNPkY_bin.7.MAG whole-genome shotgun database (BioProject ID:PRJEB47281). These contigs do not overlap; however, we assumed that they belong to the genome of the putative phage Wonh. Indeed, we cannot exclude that these contigs are genome fragments of two related *Pseudomonas* phages. Regardless of whether these contigs belong to one or two phage genomes, their nucleotide identities with the PseuGes_254 genome (NI ~ 66%) do not allow them to be attributed to the same genus as PseuGes_254.

Notably, the PseuGes_254 genome has a similar arrangement of genes in genetic clusters, or genomic synteny, to those of *Pseudomonas* phages belonging to the subfamilies *Gorskivirinae* and *Skurskavirinae* as well as the genus *Nankokuvirus* (Figure 6 and Figure 7). Members of both subfamilies have terminally redundant genomes with a size of approximately 94–96 kilobases (kb) and a GC-content of 46.8‒49.5%. According to the ICTV-taxonomy (https://ictv.global/msl, accessed on 4 July 2024), the subfamily *Skurskavirinae* consists of two genera: *Baldwinvirus* and *Pakpunavirus*. The genomes of phages from these genera encode approximately 190 proteins and 17 tRNA genes. The subfamily *Gorskivirinae* is more diverse than *Skurskavirinae*, comprising five genera: *Dilongvirus*, *Shenlongvirus*, *Kremarvirus*, *Otagovirus*, and *Flaumdravirus*. The genomes of these viruses contain approximately 170 ORFs corresponding to proteins, and from 9 to 21 genes for tRNAs. At the same time, the genomes of phages from the genus *Nankokuvirus* are smaller in size (~86–88 kb) and have a higher content of GC-nucleotides (51–54%). They contain fewer putative ORFs (150–160) and encode only 2–3 tRNAs. According to these parameters, the PseuGes_254 genome (95,072 bp, the GC content is 43%, 183 putative protein-encoding and 12 tRNA genes) is more similar to the genomes of phages from the *Gorskivirinae* and *Skurskavirinae* subfamilies than to the *Nankokuvirus* genus. However, an important feature of PseuGes_254 and Wonh phages, which distinguishes them from members of the *Gorskivirinae* and *Skurskavirinae* subfamilies and the *Nankokuvirus* genus, is the presence of a large genomic inversion (~20,000 bp). This inverted region contains genes that encode essential proteins involved in nucleic acid metabolism (Figure 3), so it probably contains all the necessary regulatory elements for the expression of these genes. As no repeats or any structures of mobile elements were found at the end of this fragment, we can not explain the occurrence of such an inversion in the genome of a lytic phage.

Phages of the *Skurskavirinae* subfamily and *Nankokuvirus* genus are able to infect *P. aeruginosa* strains, according to GenBank annotation data (https://www.ncbi.nlm.nih.gov/nucleotide/, accessed on 4 July 2024). For most members of the *Gorskivirinae* subfamily, with the exception of *Pseudomonas* phage PPSC2 (NC_073678), *P. syringae* was identified as the host. Apparently, this large group of *Pseudomonas* phages, including the *Gorskivirinae*, *Skurskavirinae*, and *Nankokuvirus*, had a common ancestor in the past. Then, they diverged and adapted to different species and strains of *Pseudomonas*. When more *P. gessardii*-specific phages and phages infecting other environmental *Pseudomonas* spp. will be discovered, these phages could be grouped into a separate taxonomic unit similar to the *Gorskivirinae* subfamily, containing phages specific to various *Pseudomonas* species.

In conclusion, the PseuGes_254 phage is a strictly lytic bacteriophage that is specific to *P. gessardii* proteolytic strains and possibly has the potential to inhibit the growth of food-spoiling *Pseudomonas*. Given the ability of PseuGes_254 to lyse *P. gessardii* proteolytic strains at low temperature, this phage or its enzymes should be further investigated for possible use in the dairy industry.

## Figures and Tables

**Figure 1 viruses-16-01561-f001:**
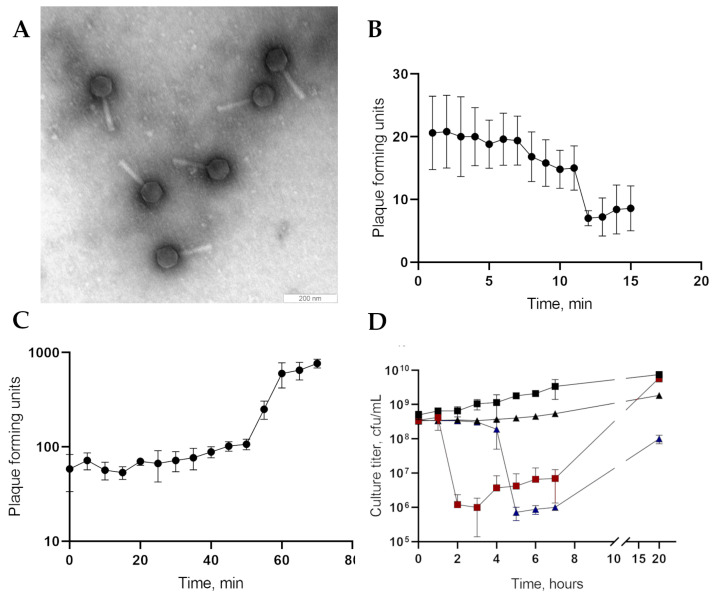
Phage PseuGes_254 characteristics. (**A**) Electron micrograph of phage PseuGes_254. (**B**) Phage adsorption assay. (**C**) One-step growth experiments. (**D**) Multistep bacterial lytic curves for the host bacterium *P. gessardii* CEMTC 4637, infected with phages, are shown. The growth at 25 °C is indicated by boxes (red boxes for the phage-infected culture and black ones for the control culture), and the growth at 10 °C is shown by triangles (blue triangles for the phage-infected culture and black ones for the control culture). The bars show standard deviations for each point.

**Figure 2 viruses-16-01561-f002:**
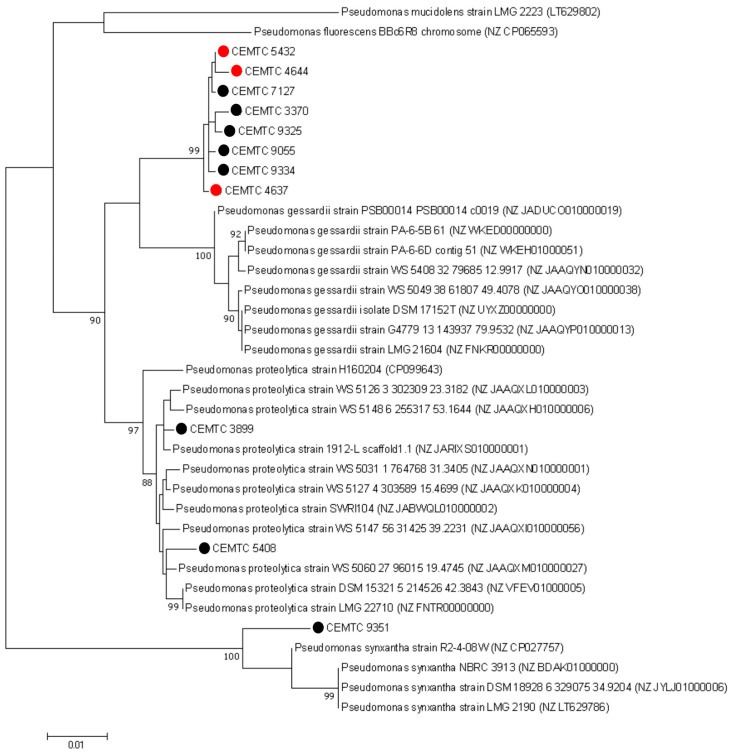
Phylogenetic analysis of *P. gessardii* subgroup isolates. Concatenated gene sequences of rpoB, rpoD, and gyrB were used. Sequences were aligned using the ClustalW algorithm. Phylogenetic trees were constructed using the maximum likelihood (ML) method based on the JTT matrix-based LG model in MEGA 11.0 with 1000 bootstrap replicates. Bacterial strains sensitive to the PseuGes_254 were marked with red circles, other investigated Pseudomonas isolates are marked with black circles.

**Figure 3 viruses-16-01561-f003:**
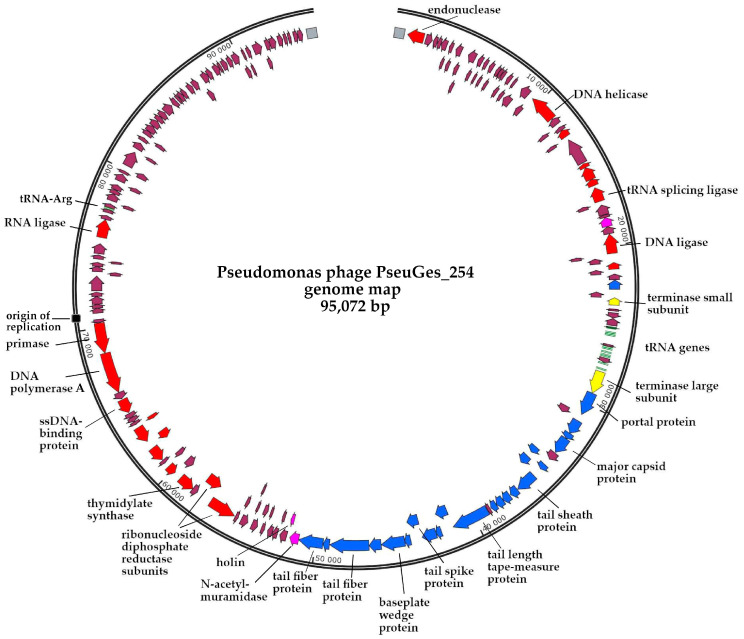
Pseudomonas phage vB_PseuGes_254 genome map. Genes encoding structural proteins are marked with blue arrows; genes corresponding to nucleic acids metabolism are marked with red; terminase subunits are yellow; genes encoding proteins of lysis cassette are rose; tRNA genes are green; other genes encoding hypothetical proteins are brown; DTRs and origin of replication are marked with grey and black boxes.

**Figure 4 viruses-16-01561-f004:**
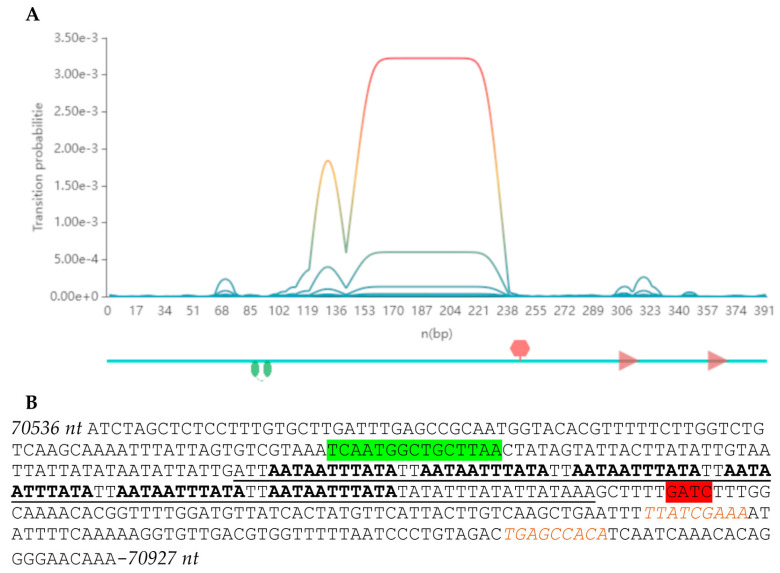
Phage origin prediction in the PseuGes_254 genome Ori-Finder software [52]. (**A**) The scheme of the putative origin of replication: CtrA binding motif is marked with green circles, GATC-region is marked with red hexagon, DnaA boxes are marked with brown arrows. (**B**) 70,536–70,927 nt sequence with highlighted elements of a putative origin: CtrA binding motif is marked with green, AT-containing region is underlined, AT-repeats are marked with bold, GATC-region is marked with red, DnaA boxes are marked with brown italics.

**Figure 5 viruses-16-01561-f005:**
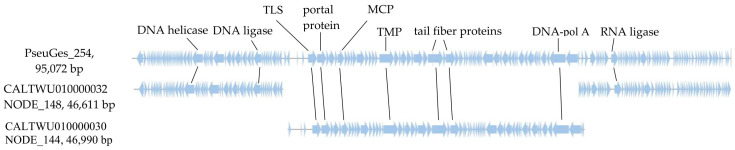
Alignment of the nucleotide sequences of the PseuGes_254 genome and two contigs from BioProject ID:PRJEB4728, namely NODE_144_length_46990_cov_333.200810 (CALTWU010000030) and NODE_148_length_46611_cov_317.059283 (CALTWU010000032). Several ORFs encoding signature proteins are indicated: TLS—terminase large subunit, MCP—major capsid protein, TMP—tape measure protein, DNA-pol A—DNA polymerase A.

**Figure 6 viruses-16-01561-f006:**
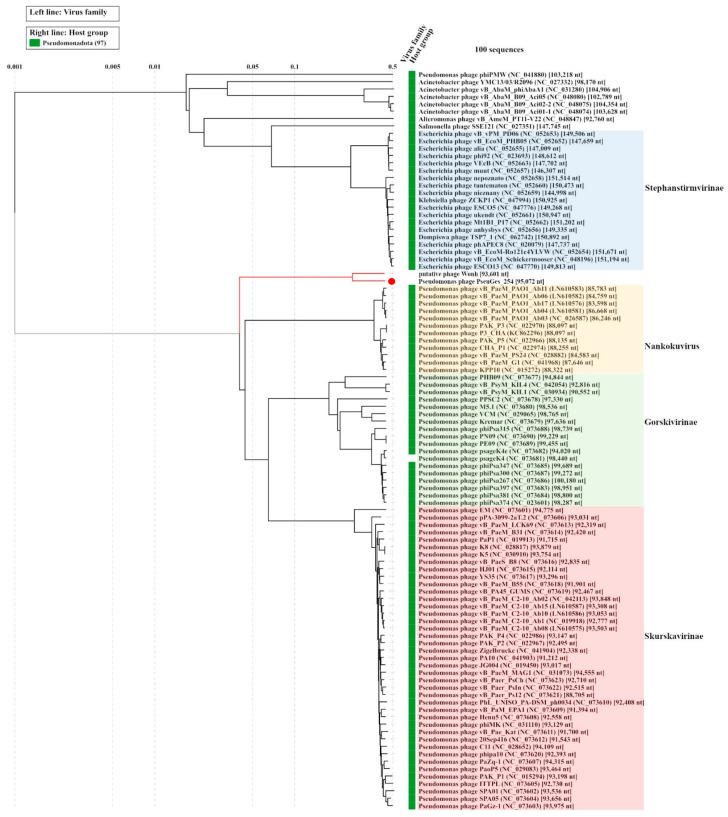
ViPTree analysis for phage PseuGes_254 and related phages. PseuGes_254 phage and putative phage Wonh are marked with red lines, PseuGes_254 is marked with red circle.

**Figure 7 viruses-16-01561-f007:**
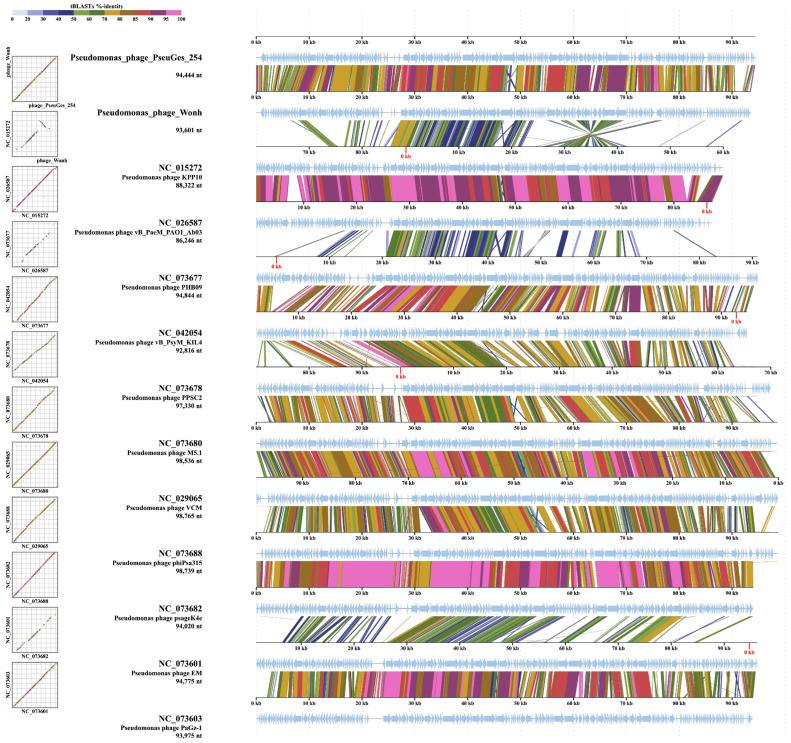
A pairwise comparison of the PseuGes_254 genome and a number of the most similar *Pseudomonas* phages was performed using the ViPTree tool.

**Figure 8 viruses-16-01561-f008:**
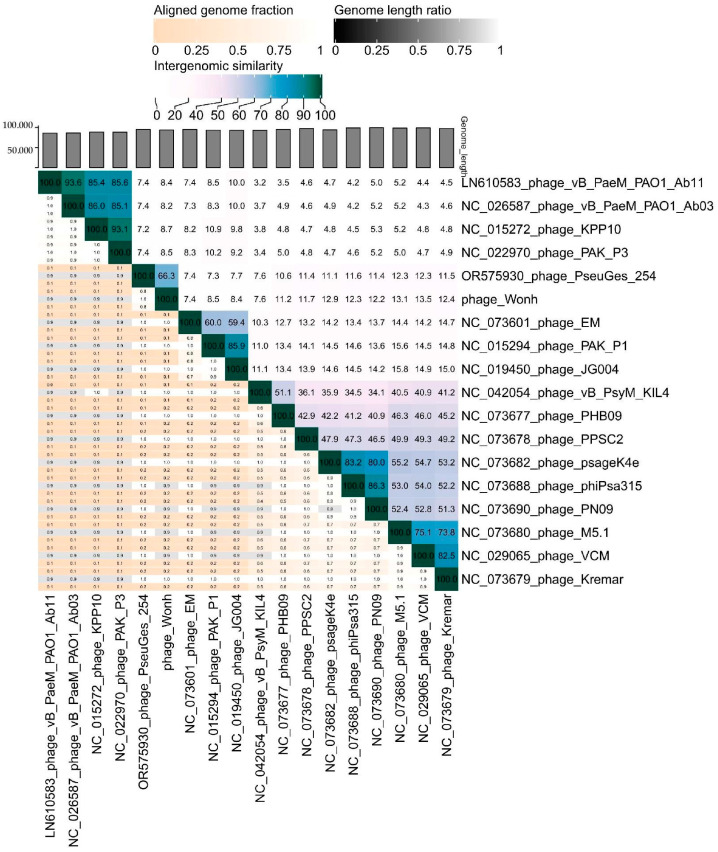
VIRIDIC heatmap indicating intergenomic similarity between the genomes of PseuGes_254 and related *Pseudomonas* phages.

**Table 1 viruses-16-01561-t001:** Bacterial strains sensitive to PseuGes_254 phage.

No.	Species (16S rRNA Gene (GenBank ID)	CEMTC Number	Isolation Source	Date of Isolation	Relative Efficiency of Plating (EOP) ^1^/Phage Titer, BOE/mL
1	*P. gessardii* (ON838145)	4637 *	Chemal river, Altai Republic	26 January 2022	Not applicable/3.7 × 10^10^
2	*P. gessardii* (OQ834588)	4644	Chemal river, Altai Republic	29 January 2022	Low/8 × 10^6^
3	*P. gessardii* (PP348772)	5432	Upper Inegen river, Altai Republic	7 May 2022	Low/1.4 × 10^7^

*—host strain, ^1^ the EOP value = phage titer on test strain/phage titer on host strain. EOP values of >1 were ranked as “high” efficiency; 0.2–1 as “medium” efficiency; 0.001–0.2 as “low” efficiency.

## Data Availability

The PseuGes_254 complete genome was deposited in the NCBI GenBank database (accession number OR575930).

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
