# Peer review of "The First Pseudomonas Phage vB_PseuGesM_254 Active against Proteolytic Pseudomonas gessardii Strains"

_viruses, 2024, doi:10.3390/v16101561_

Round 1

Reviewer 1 Report

Comments and Suggestions for Authors

The manuscript submitted by the authors, though encompassing a significant amount of data, requires substantial editing and revising prior to publication. Each section must be scrutinized and in particular the Materials & Methods section. The manuscript leaves out a considerable amount of necessary information. The paper also needs to be read and edited by a person who thoroughly understands the English language, as there are numerous grammatical errors (e.g., missing articles), and words used incorrectly. I’ve noted some but not all occurrences in the comments. Nevertheless, the entire text needs to be reexamined. The discussion also suffers in that it is a not an adequate interpretation of the results and its significance, as well as presenting no concluding analysis of why this all matters. Ultimately, there is no decisive evaluation of why all their hard work is meaningful and how the information can be used for subsequent research and/or relates to present-day needs of the science community or general population. The conclusion itself comprises two sentences. Which for the amount of work the authors accomplished, the results they discovered and the potential this data might hold, is simply not enough to accurately synopsize the manuscript. All of these issues currently make this paper challenging at best to support for publication. However, once addressed, I trust the manuscript will make an brilliant contribution to the bacteriophage community and beyond. Please see specific comments below.

Page 1, line 14-15. The authors state that “phages infecting environmental Pseudomonas spp. have not been well studied…” I suggest the authors attempt a more thorough and diligent search. A simple Google Scholar search turned up a considerable number of scientific articles referencing Pseudomonas spp. and bacteriophages. Hence, this statement sincerely confuses me and will likely confuse many of the readers of this article.

Page 1, line 33. The word “validly” is improperly used in this statement, as it implies that other published species are irrelevant, false and useless. This is a strong statement that surely the authors do not wish to lob at other scientists’ work. Please find a more appropriate word to convey what you mean.

Page 2, lines 46-49. This sentence is not clear, please revise to make it more comprehensible.

Page 2, lines 61-62. Same as first comment. The authors have also not indicated “why” these phages have not been studied well if that is what they are asserting. There needs to be a reason for the dearth of research.

Page 2. Line 80. By “one gram of sample” is this a sample of sediment or bacteria? If this was from sediment, why was the sediment not agitated first to remove all potentially adhered to Pseudomonas?

Page 2. Line 82. What was the temperature at which centrifugation was performed?

Page 2. Line 83. What were the tenfold dilutions that were plated?

Page 2. Lines 84-86. Please clarify what “independently passaged three times under the same conditions” means. At what points in the previous conditions were the isolates processed? This is quite unclear.

Page 2, Line 86. As an example of grammar errors, “To identify bacterial isolate, a fragment…” should instead be “To identify a bacterial isolate, a fragment…” Please review the text for further grammatical errors such as this.

Page 2. Line 86-87. There is missing information here. How was the bacterial DNA obtained? How was it purified and quantified? How was the 16s amplified? What were the parameters? How as it prepared for sequencing? How was it sequenced? How were the raw data evaluated, read and finally assembled? The current statement written is insufficient to convey the methods used for this portion of the methods and needs serious attention. If mentioned here, the information needs to be here as well.

Page 2. Line 88. What was the percentage of Nutrient Broth used?

Page 2. Lines 91-93. How was the strain deposited? Was it the DNA? A lyophilized sample? This needs much more clarification.

Page 2, Line 95. The text in the parentheses prior to “Table S2” is unnecessary.

Page 2. Line 96-97. How was this done? DNA extraction, purification, PCR, etc. Need references and data.

Page 3. Line 98. Same comments about missing method data regarding this process.

Page 3. Line 100-102. What percentage milk agar? What ratio milk containing plates? Need reference.

Page 3. Line 107. How was the sample clarified? Is this a 0.22uM pore size membrane filter or other type of filter?

Page 3. Line 108-109. When were the top agar plates made and how were they made?

Page 3. Line 111. How long was “some hours”? 2, 10, 24?

Page 3. Line 117. What percentage Nutrient Broth and how did you determine the OD? Machine, manufacturer.

Page 3. Line 119-120. How long is “several hours” and at what temperature was this done?

Page 3. Line 121, and 134. Centrifuge speed, time, temperature and machine manufacturer are all missing.

Page 3. Section 2.3. Were the P. gessardii cells being agitated during exponential growth?

Page 3. Line 135 and 139. How much were the aliquots?

Page 3. Line 139. Please specify what mixture you are referring to.

Page 3. Line 139-140. What were the “appropriate dilutions”? Please specify.

Page 3. Line 145-146. Please reword, this is unclear.

Page 4. Section 2.4. Was the phage suspension ever subjected to a CsCl density gradient purification or other process to ensure the sample was adequately purified prior to EM analysis?

Page 4. Section 2.5. This section is a prime example of what the previous M&M sections should have contained. It has ample information, references and doesn’t leave out vital data. The authors can use this as their guide to correct the other deficient sections within.

Page5. Figure 1.D. the graph needs to be colour coded so that there is a distinction between the points annotated. As is now, it is quite problematic to read. Also, in the figure legend the authors need to address what the bars symbolize in each graph, is it standard deviations? Please include all data.

Page 7. Figure 2. Please colour the dots as they are difficult to distinguish from each other and particularly when some of the numbers appear as potential icons.

Page 8. Figure 4. A prospective way to make this genome map much easier to read would be to colour code the text to match the genes. Such as typing “tRNA” in green and “terminase subunits” in yellow, and the others as well. This would make it easier to read and quickly locate information on the map. Also, the GC content  could also be included on the genome map.

Figures 6 and 7. Both of these include text too small to be legible. Perhaps enlarge each to a full page so that the information can be distinguished from merely indistinct lines to clear text. Either that or move to the Supplementary section for a larger space.

Page 12, Lines 361-361. The authors assert that the bacteriophage discovered and studied here in this manuscript has a strong lytic activity. Why then are there no specific sections directing the reader to their objective of this statement? The data about lytic activity is nestled in M&M section 2.3, without any direct call out or bolding that this is a big part of their report. In the Results, a total of two sentences describe the Figure 1D that presents the lytic activity experiment. One remark among the gene functions, and then the Discussion assertion that the phage has strong lytic activity and is strictly lytic. If this is indeed a prominent point to the manuscript, it would serve the paper better to be emphasized, enhanced and given the distinction it merits.

Supplementary Sections: There are no figure legends accompanying any of the figures in the supplementary section. This is unhelpful for the reader and needs to be addressed. The tables do have legends, albeit a bit abbreviated and could use a review for increasing their substance.

Comments on the Quality of English Language

As mentioned already in the review comments, the English language needs addressed.

Author Response

Reviewer 1

We would like to thank the Reviewer for his/her valuable comments, which we believe help us to improve our manuscript. Our corrections according to the Reviewer 1 comments were   marked with green in the text.

The manuscript submitted by the authors, though encompassing a significant amount of data, requires substantial editing and revising prior to publication. Each section must be scrutinized and in particular the Materials & Methods section. The manuscript leaves out a considerable amount of necessary information. The paper also needs to be read and edited by a person who thoroughly understands the English language, as there are numerous grammatical errors (e.g., missing articles), and words used incorrectly. I’ve noted some but not all occurrences in the comments. Nevertheless, the entire text needs to be reexamined. The discussion also suffers in that it is a not an adequate interpretation of the results and its significance, as well as presenting no concluding analysis of why this all matters. Ultimately, there is no decisive evaluation of why all their hard work is meaningful and how the information can be used for subsequent research and/or relates to present-day needs of the science community or general population. The conclusion itself comprises two sentences. Which for the amount of work the authors accomplished, the results they discovered and the potential this data might hold, is simply not enough to accurately synopsize the manuscript. All of these issues currently make this paper challenging at best to support for publication. However, once addressed, I trust the manuscript will make an brilliant contribution to the bacteriophage community and beyond. Please see specific comments below.

Page 1, line 14-15. The authors state that “phages infecting environmental Pseudomonas spp. have not been well studied…” I suggest the authors attempt a more thorough and diligent search. A simple Google Scholar search turned up a considerable number of scientific articles referencing Pseudomonas spp. and bacteriophages. Hence, this statement sincerely confuses me and will likely confuse many of the readers of this article.

We agree with the Reviewer that there have been a number of studies on environmental Pseudomonas phages. Moreover, some of those were cited in our manuscript [29-36]. The phrase was removed from the text to clarify this point.

Page 1, line 33. The word “validly” is improperly used in this statement, as it implies that other published species are irrelevant, false and useless. This is a strong statement that surely the authors do not wish to lob at other scientists’ work. Please find a more appropriate word to convey what you mean.

We didn't mean to hurt any researcher's feelings. The term “validly published” was a citation taken from the internet resource, https://lpsn.dsmz.de/genus/pseudomonas, access date June 03, 2024. This term is commonly used and refers to bacterial species that have been approved by a taxonomic committee.

Page 2, lines 46-49. This sentence is not clear, please revise to make it more comprehensible.

 Was done, Lines 47-50

Page 2, lines 61-62. Same as first comment. The authors have also not indicated “why” these phages have not been studied well if that is what they are asserting. There needs to be a reason for the dearth of research.

The confusing phrase has been removed from the text. We meant that most of the Pseudomonas phages (~850), which can be accessed at https://www.ncbi.nlm.nih.gov/genbank on July 1st, 2024, were isolated using Pseudomonas aeruginosa as a host. And only a small number of Pseudomonas phages (37) are known to infect host bacteria from the P. fluorescens group. Probably, the P. aeruginosa phages have been isolated as a potential therapeutic agent, which is being developed due to the increased antibiotic resistance of this organism and its significant importance in clinical practice.

Please, find Lines 65-67

Page 2. Line 80. By “one gram of sample” is this a sample of sediment or bacteria? If this was from sediment, why was the sediment not agitated first to remove all potentially adhered to Pseudomonas?

Was corrected. Line 86-88. “One gram of the sediment sample was suspended… and agitated…”

Page 2. Line 82. What was the temperature at which centrifugation was performed?

 Was added. Line 89

Page 2. Line 83. What were the tenfold dilutions that were plated?

 Was corrected. Line 91

Page 2. Lines 84-86. Please clarify what “independently passaged three times under the same conditions” means. At what points in the previous conditions were the isolates processed? This is quite unclear.

The method description was corrected. Lines 92-97

Page 2, Line 86. As an example of grammar errors, “To identify bacterial isolate, a fragment…” should instead be “To identify a bacterial isolate, a fragment…” Please review the text for further grammatical errors such as this.

Was done. Line 98 

Page 2. Line 86-87. There is missing information here. How was the bacterial DNA obtained? How was it purified and quantified? How was the 16s amplified? What were the parameters? How as it prepared for sequencing? How was it sequenced? How were the raw data evaluated, read and finally assembled? The current statement written is insufficient to convey the methods used for this portion of the methods and needs serious attention. If mentioned here, the information needs to be here as well.

We added a description of routine 16S rRNA sequencing and included a reference. Lines 99-126

Page 2. Line 88. What was the percentage of Nutrient Broth used?

Was added. Line 128.

Page 2. Lines 91-93. How was the strain deposited? Was it the DNA? A lyophilized sample? This needs much more clarification.

Was added. Lines 134-135

 Page 2, Line 95. The text in the parentheses prior to “Table S2” is unnecessary.

Was removed

Page 2. Line 96-97. How was this done? DNA extraction, purification, PCR, etc. Need references and data.

Was added. Lines 99-126

Page 3. Line 98. Same comments about missing method data regarding this process.

Was added 

Page 3. Line 100-102. What percentage milk agar? What ratio milk containing plates? Need reference.

 Was corrected. Lines 137-141

Page 3. Line 107. How was the sample clarified? Is this a 0.22uM pore size membrane filter or other type of filter?

Yes, the sample was sterilized using filtration. Please, find lines 145-148 “The supernatant obtained after clarification of the sample was filtered through a 0.22 µM filter (Millipore, Guyancourt, USA)…”

Page 3. Line 108-109. When were the top agar plates made and how were they made?

 Was re-written. Lines 148-153

Page 3. Line 111. How long was “some hours”? 2, 10, 24?

 Was corrected to 16 hours. Line 153

Page 3. Line 117. What percentage Nutrient Broth and how did you determine the OD? Machine, manufacturer.

Was added. Lines 160-161

Page 3. Line 119-120. How long is “several hours” and at what temperature was this done?

 Was corrected. Line 163

Page 3. Line 121, and 134. Centrifuge speed, time, temperature and machine manufacturer are all missing.

 Was corrected. Line 165

Page 3. Section 2.3. Were the P. gessardii cells being agitated during exponential growth?

 Was added. Lines 169-175

Page 3. Line 135 and 139. How much were the aliquots?

 Were added.Line 174

Page 3. Line 139. Please specify what mixture you are referring to.

 Was corrected. Line 173,178

Page 3. Line 139-140. What were the “appropriate dilutions”? Please specify.

Was corrected. Lines 182-196

Page 3. Line 145-146. Please reword, this is unclear.

Was corrected. Lines 182-196

Page 4. Section 2.4. Was the phage suspension ever subjected to a CsCl density gradient purification or other process to ensure the sample was adequately purified prior to EM analysis?

No, the phage preparation was not purified using CsCl density gradient. The phage-containing supernatant was sterilized through 0.22 mkM filter and examined for phage particles morphology in EM analysis.

Page 4. Section 2.5. This section is a prime example of what the previous M&M sections should have contained. It has ample information, references and doesn’t leave out vital data. The authors can use this as their guide to correct the other deficient sections within.

We thank the Reviewer for these comments.

Page5. Figure 1.D. the graph needs to be colour coded so that there is a distinction between the points annotated. As is now, it is quite problematic to read. Also, in the figure legend the authors need to address what the bars symbolize in each graph, is it standard deviations? Please include all data.

The 1D graph was coloured, and Figure legend was corrected.

Page 7. Figure 2. Please colour the dots as they are difficult to distinguish from each other and particularly when some of the numbers appear as potential icons.

 Was done

Page 8. Figure 4. A prospective way to make this genome map much easier to read would be to colour code the text to match the genes. Such as typing “tRNA” in green and “terminase subunits” in yellow, and the others as well. This would make it easier to read and quickly locate information on the map. Also, the GC content  could also be included on the genome map.

We appreciate the reviewer's input, but the tool we used to create this map does not allow for the calculation of GC-content. In addition, we believe that yellow text is difficult to read, and therefore, we would prefer not to make changes in the figure.

Figures 6 and 7. Both of these include text too small to be legible. Perhaps enlarge each to a full page so that the information can be distinguished from merely indistinct lines to clear text. Either that or move to the Supplementary section for a larger space.

Both figures were prepared in accordance with the journal's guidelines for authors (TIFF, 300 dpi). These figures can be enlarged in the "DOC" version of the manuscript, and the details will be clear.

Page 12, Lines 361-361. The authors assert that the bacteriophage discovered and studied here in this manuscript has a strong lytic activity. Why then are there no specific sections directing the reader to their objective of this statement? The data about lytic activity is nestled in M&M section 2.3, without any direct call out or bolding that this is a big part of their report. In the Results, a total of two sentences describe the Figure 1D that presents the lytic activity experiment. One remark among the gene functions, and then the Discussion assertion that the phage has strong lytic activity and is strictly lytic. If this is indeed a prominent point to the manuscript, it would serve the paper better to be emphasized, enhanced and given the distinction it merits.

We appreciate the reviewer's comment regarding the use of the term "lytic activity". This is a biological property that can be determined for new phages, and it is a commonly measured parameter for phages. To avoid misinterpretation of the meaning and purpose of our article, we have chosen to remove the word "strong" from the text.

Supplementary Sections: There are no figure legends accompanying any of the figures in the supplementary section. This is unhelpful for the reader and needs to be addressed. The tables do have legends, albeit a bit abbreviated and could use a review for increasing their substance.

Supplementary data was prepared in accordance with the journal's guidelines for authors. Therefore legends were presented at the end of the manuscript in the section, which is called

“Supplementary Materials”

Comments on the Quality of English Language

Thanks to the Reviewer, we checked the quality of the text.

As mentioned already in the review comments, the English language needs addressed.

Reviewer 2 Report

Comments and Suggestions for Authors

The paper is very worthy, of great interest to the scientific community. The authors have carefully worked through the material, obtained interesting results, the paper should certainly be published.

To improve the quality of the paper, I recommend to correct small errors and supplement in accordance with the comments.

Line 110

I suggest replacing “after clarification” with “after low-speed centrifugation of the sediment sample”

Line 140 it is written that an exponentially growing culture was taken in the experiment, but in Figure 1D it can be seen that at the start the culture had a titer of 108 - 109 and it did not increase with time in the control “exponentially”. The culture appears to be in stationary phase. 

Line 163-172 “The purification of PseuGes........... purified using ethanol precipitation”

Line 168, 170 “ mkg/mL ” I suggest replacing with µg/mL.

Line 177 – how was the assembly checked, Bandage, CheckV used?

Line 180 – specify which program was used to obtain ORF.

Line 210 “two hours” – however, the graph shows that 2 hours at 25°C and 4 hours at 10°C.

Fig 1 on the y-axis of Fig. 1B and 1C better specify plaque forming units per cell, on Fig. 1D better specify CFU/1 mL.

Fig 2 it is better to specify that white circles indicate strains not sensitive to PseuGes_254 phage

I also suggest using geNomad for taxonomic annotation.

Have you tried using phyre2 (http://www.sbg.bio.ic.ac.uk/) to annotate hypothetical proteins as well as the PHROG database?

I recommend reading 10.1089/phage.2021.0015 and doi.org/10.1089/phage.2021.0013

Author Response

Reviewer2

The paper is very worthy, of great interest to the scientific community. The authors have carefully worked through the material, obtained interesting results, the paper should certainly be published.

We are grateful to the Reviewer for the very interesting and useful comments, especially for the links to bioinformatic tools. We made corrections in the text according to Reviewer comments and they are marked with blue.

To improve the quality of the paper, I recommend to correct small errors and supplement in accordance with the comments.

Line 110

I suggest replacing “after clarification” with “after low-speed centrifugation of the sediment sample”

Was done. Line 145

Line 140 it is written that an exponentially growing culture was taken in the experiment, but in Figure 1D it can be seen that at the start the culture had a titer of 108 - 109 and it did not increase with time in the control “exponentially”. The culture appears to be in stationary phase.

We thank the Reviewer for this comment. Yes, it was a mistake in methods. The culture had a high titer. We corrected this point in methods. Lines 183-185.

Line 163-172 “The purification of PseuGes........... purified using ethanol precipitation”

Was corrected. Lines 212, 220-222

Line 168, 170 “ mkg/mL ” I suggest replacing with µg/mL.

Was done. Line 216

Line 177 – how was the assembly checked, Bandage, CheckV used?

We would like to thank the reviewer for their comment. We did not use these tools in this study. It was not a metagenomic sample but a purified phage DNA, and the phage DNA contig was assembled using SPades. The resulting contig was circular and had a high average coverage of 216, which indicates that it is likely to be a complete genome for the phage. Additionally, the PhageTerm tool was successfully used to find the termini of the genome.

 Line 180 – specify which program was used to obtain ORF.

Rapid Annotation Subsystem Technology (RAST) v.2.0 [47] (https://rast.nmpdr.org, ac-cessed on August 12, 2023) was used to find and annotate the putative ORFs. Next step, other resources were used to confirm or clear this annotation. Please, find Lines 228-234

Line 210 “two hours” – however, the graph shows that 2 hours at 25°C and 4 hours at 10°C.

Was corrected. Lines 259-260

Fig 1 on the y-axis of Fig. 1B and 1C better specify plaque forming units per cell, on Fig. 1D better specify CFU/1 mL.

“CFU” was corrected to cfu/ml

Fig 2 it is better to specify that white circles indicate strains not sensitive to PseuGes_254 phage

Thanks, the Figure and its legend were corrected.

I also suggest using geNomad for taxonomic annotation.

Thanks, we will try this tool.

Have you tried using phyre2 (http://www.sbg.bio.ic.ac.uk/) to annotate hypothetical proteins as well as the PHROG database?

Thanks, it is a very interesting opportunity. We’ll try this tool for annotation.

I recommend reading 10.1089/phage.2021.0015 and doi.org/10.1089/phage.2021.0013

We are grateful for the Reviewer for these useful links.  We read the last paper (doi.org/10.1089/phage.2021.00) and found that we apply the similar pipeline to annotate our phage genomes. Please, find Materials and Methods, 2.5. Complete Genome Sequencing and Analysis.

Round 2

Reviewer 1 Report

Comments and Suggestions for Authors

The authors have taken to heart the recommendations and comments for their manuscript. The revised abstract is in particular much improved. The additional data incorporated into the M&M section vastly enhances this section and provides the essential information for all readers who wish to know the particulars and understand how the information was obtained. Fantastic job. I would have liked to have seen a bit more extrapolation for the manuscript conclusion, however.

Furthermore, I did eventually locate the figure legends for the Supplementary section. I must say, this is a misstep on the journal’s part to have the authors assemble the information into a single paragraph, sentence after sentence, without breaks or indication of separate figures (bold text, etc.). It does not facilitate reviewing by virtually intermingling the text with the acknowledgments, funding and other information. I would recommend always including a page of the figure legends, as perhaps bullet points, on a separate page with the Supplementary material files. It facilitates the review process tremendously.